# Molecular Mechanisms: Connections between Nonalcoholic Fatty Liver Disease, Steatohepatitis and Hepatocellular Carcinoma

**DOI:** 10.3390/ijms21041525

**Published:** 2020-02-23

**Authors:** Tatsuo Kanda, Taichiro Goto, Yosuke Hirotsu, Ryota Masuzaki, Mitsuhiko Moriyama, Masao Omata

**Affiliations:** 1Division of Gastroenterology and Hepatology, Department of Medicine, Nihon University School of Medicine, 30-1 Oyaguchi-kamicho, Itabashi-ku, Tokyo 173-8610, Japan; kanda2t@yahoo.co.jp (T.K.); masuzaki.ryota@nihon-u.ac.jp (R.M.); moriyama.mitsuhiko@nihon-u.ac.jp (M.M.); 2Lung Cancer and Respiratory Disease Center, Yamanashi Central Hospital, 1-1-1 Fujimi, Kofu, Yamanashi 400-8506, Japan; 3Genome Analysis Center, Yamanashi Central Hospital, Yamanashi 400-8506, Japan; hirotsu-bdyu@ych.pref.yamanashi.jp (Y.H.); m-omata0901@ych.pref.yamanashi.jp (M.O.); 4The University of Tokyo, 7-3-1 Hongo, Bunkyo-ku, Tokyo 113-8655, Japan

**Keywords:** dysbiosis, HCC, HSD17B13, NAFLD, NASH, metabolic syndrome, microbiota, obesity, PNPLA3

## Abstract

Nonalcoholic fatty liver disease (NAFLD), including nonalcoholic steatohepatitis (NASH), causes hepatic fibrosis, cirrhosis and hepatocellular carcinoma (HCC). The patatin-like phospholipase-3 (PNPLA3) I148M sequence variant is one of the strongest genetic determinants of NAFLD/NASH. PNPLA3 is an independent risk factor for HCC among patients with NASH. The obesity epidemic is closely associated with the rising prevalence and severity of NAFLD/NASH. Furthermore, metabolic syndrome exacerbates the course of NAFLD/NASH. These factors are able to induce apoptosis and activate immune and inflammatory pathways, resulting in the development of hepatic fibrosis and NASH, leading to progression toward HCC. Small intestinal bacterial overgrowth (SIBO), destruction of the intestinal mucosa barrier function and a high-fat diet all seem to exacerbate the development of hepatic fibrosis and NASH, leading to HCC in patients with NAFLD/NASH. Thus, the intestinal microbiota may play a role in the development of NAFLD/NASH. In this review, we describe recent advances in our knowledge of the molecular mechanisms contributing to the development of hepatic fibrosis and HCC in patients with NAFLD/NASH.

## 1. Introduction

Nonalcoholic fatty liver disease (NAFLD), including nonalcoholic steatohepatitis (NASH), is a leading cause of liver-related morbidity and mortality in the United States [1]. NASH presents with increased hepatic fibrosis and progresses to end-stage liver disease [2]. Patients with NASH also have a potential risk of developing hepatocellular carcinoma (HCC) [3]. Clinicians should be aware of the possibility of an occurrence of HCC in patients with NASH and the importance of prevention of NAFLD, including NASH, in adults as well as adolescents [4].

Steatohepatitis is a morphological pattern of liver injury that may be seen in both NAFLD and alcoholic liver disease [5]. Although NASH is closely associated with obesity, diabetes mellitus and related abnormalities, such as dyslipidemia and insulin resistance, as well as the use of certain drugs, its natural history is still poorly understood [5,6,7]. The concept of NASH was first established in the context of pathology studies of the liver [5,6]. However, it may be important to study this disease from the standpoint of genetic alterations.

In this article, we focus on the molecular mechanisms shared by NAFLD, including NASH and HCC development, and we discuss recent findings on the molecular mechanisms underlying NAFLD, NASH and HCC.

## 2. Definition of NASH

Ludwig et al. first reported that NASH is a disease that histologically mimics alcoholic liver disease (ALD) and may progress to cirrhosis [8]. The liver biopsy specimens were shown to exhibit striking fatty changes with evidence of lobular hepatitis, focal necroses with mixed inflammatory infiltrates and Mallory bodies. In most of the specimens studied, fibrosis of the liver was evident, and in some patients, cirrhosis had been diagnosed [8]. Most of the patients were found to be moderately obese with diabetes mellitus and cholelithiasis [8]. Exclusions were made for patients with known alcohol use in excess of 20 g per day, as confirmed by the referring physician and a family member [5]. A deposition of collagen was histologically observed in zone 3 and in the perisinusoidal area, and bridging peri-sinusoidal fibrosis was also observed [5].

Hepatic steatosis is caused by both ALD and NAFLD. NAFLD is observed in patients with obesity, diabetes mellitus, jejunum/ileum anastomosis, Weber–Christian disease, endocrine diseases such as Cushing syndrome, hypothyroidism, growth hormone deficiency or polycystic ovary syndrome, as well as in cases of severe malnutrition, such as Kwashiorkor, Crohn’s disease, short bowel syndrome or anorexia nervosa, and also in patients after gastroplasty, small bowel resection, pituitary tumor surgery, or treatment with certain drugs, such as amiodarone, tamoxifen, corticosteroids or antiviral drugs [9,10].

## 3. Pathophysiology of NASH

Obesity, metabolic syndrome and hepatocyte apoptosis are associated with NAFLD and NASH [11,12,13]. Both innate and acquired immunity mediate the development of insulin resistance and NASH [14]. Recent studies revealed that intestinal bacteria are also involved in the development of NASH [15,16,17]. Thus, many factors contribute to the development of these two conditions (Table 1).

### 3.1. Obesity

Younossi et al. examined 8,515,431 patients from 22 countries in 85 studies and showed the global prevalence of NAFLD to be 25.2%, with the highest prevalence in the Middle East and South America and the lowest prevalence in Africa [18]. The metabolic comorbidities associated with NAFLD are hyperlipidemia (69.2%), obesity (51.3%), metabolic syndrome (42.5%), hypertension (39.3%) and type 2 diabetes (22.5%). The obesity epidemic is closely associated with the rising prevalence and severity of NAFLD. Obesity has been linked not only with simple steatosis but also with advanced diseases such as NASH [19]. Insulin resistance is a driver of the progression of NAFLD, and diabetes mellitus is highly predictive of the progression of simple steatosis to advanced liver fibrosis [20]. The hyperinsulinemia induced by peripheral insulin resistance can stimulate lipogenesis in the liver [21]. Insulin resistance can also lead to an increased release of free fatty acids by adipose tissue [22].

### 3.2. Apoptosis

Cytokeratin-18 (CK-18) fragments generated by caspase 3 are an independent predictor of NASH in patients with NAFLD [23]. Caspase-3-generated CK-18 fragments are markedly increased in the blood of patients with definitive NASH [23]. The activation of caspases and apoptosis-related molecules is frequently observed in the liver of NASH patients [12]. NAFLD progresses to NASH in response to elevated endoplasmic reticulum (ER) stress [12,24]. ER stress can drive lipogenesis and steatohepatitis via caspase-2 activation of site 1 protease [24].

Bcl-2 family members also play a role in the progression of NASH. Hepatocellular-localized endothelial nitric oxide synthase (eNOS) deletion exacerbates Western-diet-induced NASH through a decrease in PPARγ coactivator-1α, mitochondrial transcription factor A, Bcl-2-interacting protein-3 (BNIP3) and 1A/1B light chain 3B (LC3B) [25]. Activation of JNK also contributes to the apoptosis that occurs in NAFLD and NASH [12].

Tumor necrosis factor-related apoptosis-inducing ligand (TRAIL)-producing natural killer (NK) cells promote a proinflammatory environment in the early stages of NAFLD [12,26]. In patients with either NAFLD or NASH, apoptotic hepatocytes stimulate immune cells and hepatic stellate cells, which contribute to the progression of hepatic fibrosis [12]. Inflammasomes and cytokines can induce apoptosis and contribute to the development of NAFLD and NASH [12]. Oxidative stress, ER stress and autophagy also contribute to the progression of NAFLD and NASH through a modification of hepatocyte apoptosis [12]. Glucose metabolism and lipid metabolism are also involved in the progression of NAFLD and NASH [12]. Thus, apoptosis is an important factor in the pathogenesis of these two conditions.

### 3.3. Immune and Inflammatory Pathways

Intercellular adhesion molecule-1 (ICAM-1)-positive hepatocytes have been observed in NASH patients and reportedly localize in areas with microvesicular fat, although ICAM-1-positive hepatocytes have not been reported in non-NASH patients [27]. Soluble ICAM-1 (sICAM-1) is reported to be significantly higher in NASH patients than in non-NASH patients. Patients with a NAFLD activity score (NAS) greater than four have a higher number of areas of CD68-positive cells and forkhead box P3 (Foxp3)-positive cells than non-NASH patients [27]. In liver tissue with NASH, hepatocytes with microvesicular steatosis express more inflammatory markers than hepatocytes in non-NASH tissues, and in this liver tissue, an increased number of CD68+ cells, such as macrophages and Foxp3+ cells, including regulatory T cells (Tregs), is reportedly observed [27]. There are two types of macrophages, M1 (or “classically activated”) and M2 (or “alternatively activated”). They play an important role in humoral immunity and the response to pathogens, and have pro- and anti-inflammatory properties, respectively [13]. M1-skewed inflammation accompanies diet-induced obesity in the liver and adipose tissue [28]. Liver-resident macrophages and Kupffer cells are lost early after disease onset, followed by a robust infiltration of Ly-6C+ monocyte-derived macrophages in a mouse model fed a methionine-choline-deficient diet [29].

Excess iron and hepatic changes in iron metabolism are associated with the development of NAFLD and NASH [30]. Dietary iron excess leads to hepatic oxidative stress, immune cell activation, hepatocellular ballooning injury and NASH in genetically obese mice [31]. The NOD-like receptor (NLR) family pyrin domain-containing 3 (NLRP3) inflammasome is an intracellular multiprotein complex involved in the production of mature interleukin 1-beta (IL-1β) that induces metabolic inflammation [32]. The NLRP3 inflammasome also plays a crucial role in the progression of NASH in mice. Henao-Mejia et al. demonstrated that the NLRP6 and NLRP3 inflammasomes and the effector protein IL-18 negatively regulate NAFLD/NASH progression [16]. Similar to Asc−/− and caspase-1−/− mice, MCDD-fed Il18−/− animals featured a significant exacerbation of NASH severity [16].

Tim-3 plays a role as an immune checkpoint in the regulation of both adaptive and innate immune cells, including macrophages, and is involved in chronic liver diseases. Tim-3 serves as an important predictor in methionine-deficient-diet (MCD)-induced NASH by regulating reactive oxygen species (ROS) and the associated pro-inflammatory cytokine production in macrophages [33]. TNF receptor superfamily member 4 (TNFRSF4/OX40) deficiency suppresses Th1 and Th17 differentiation, and TNFRSF4 deficiency in T cells inhibits monocyte migration, antigen presentation and M1 polarization [34]. The soluble TNFRSF4 plasma levels are positively associated with NASH. TNFRSF4 is a key regulator of both intrahepatic innate and adaptive immunity and plays a role in the development of NASH [34].

These results indicate that both innate and adaptive immunity are involved in the pathogenesis of NASH.

### 3.4. Intestinal Bacteria

Wigg et al. reported that patients with NASH have a higher prevalence of small intestinal bacterial overgrowth (SIBO), according to the ^14^C-D-xylose-lactulose breath test, and they also have higher TNF-α levels than patients without NASH [35]. Nair et al. reported that higher breath ethanol concentrations are observed in obese female subjects than in nonobese females, suggesting that intestinally derived ethanol may contribute to the pathogenesis of NASH [36]. Wu et al. reported that SIBO may decrease small intestinal motility in NASH rats [37]. They also demonstrated that the effects of cidomycin significantly lowered the serum ALT, AST and TNF-alpha levels in NASH rats [37]. Although the lipopolysaccharide binding protein (LBP) levels and TLR-2 expression were found to be similar, TLR-4/MD-2 expression on CD14-positive cells was higher in the NASH patients than the control subjects [38]. NASH patients have a higher prevalence of SIBO, which is associated with an enhanced expression of TLR-4 and release of IL-8 [38]. In the small intestine and liver of the rat NASH model, it is possible that enhancement of the innate immune response through the TLR4 signal leads to increased production of TNF-α [39]. Wada et al. reported that IP-10 was higher in NAFLD models than in controls and higher in NASH models than in both NAFLD models and controls [40].

In one experiment, the intestinal mucosa barrier function was shown to have changed, resulting in the progression of nonalcoholic steatohepatitis in rats [15]. Intestinal mucosa barrier malfunction may play a role in NASH. Increased intestinal permeability and tight junction alterations are also observed in patients with NAFLD. Miele et al. demonstrated that NAFLD is associated with increased gut permeability and that this abnormality is related to the increased prevalence of SIBO [41]. F11r encodes junctional adhesion molecule A (JAM-A), and its defect leads to intestinal epithelial permeability [42]. A diet high in saturated fat, fructose and cholesterol (a high-fat and cholesterol-deficient (HFCD) diet) led to a significant increase in inflammatory microbial taxa in F11r(-/-) mice compared with control animals. The administration of oral antibiotics or sequestration of bacterial endotoxins with sevelamer hydrochloride reduced mucosal inflammation and restored normal liver histology in F11r(-/-) mice fed an HFCD diet [42]. Both the intestinal and hepatic CYP2E1 induced by binge alcohol consumption seem important in binge alcohol-mediated increased nitroxidative stress, gut leakage and endotoxemia. These factors may alter fat metabolism and inflammation, contributing to hepatic apoptosis and steatohepatitis [43].

Elinav et al. demonstrated that a deficiency of NLRP6 in mouse colonic epithelial cells resulted in the reduction of IL-18 levels and an alteration in the fecal microbiota characterized by expanded representation of the bacterial phyla Bacteroidetes (Prevotellaceae) and TM7 [43]. NLRP6 and NLRP3, both of which include ASC and caspase-1 and involve IL-18 but not IL-1R, result in the development of an altered transmissible, colitogenic intestinal microbial community [44]. Henao-Mejia et al. observed that the increased severity of NASH in Asc- and Il18-deficient mice is transmissible to cohoused wild-type mice [16]. They also demonstrated diet- and cohousing-associated changes in gut microbial ecology by 16S rRNA sequencing [16]. The levels of TLR4 and TLR9 agonists, but not TLR2 agonists, were markedly increased in the portal circulation of MCDD-fed wt (Asc−/−) and Asc−/− mice compared to wt controls, suggesting that TLR4 and TLR9 agonist efflux from the intestines of inflammasome-deficient mice or their cohoused partners through the portal circulation to the liver triggers TLR4 and TLR9 activation, that in turn results in enhanced progression of NASH [16]. Microbiota-induced subclinical colon inflammation may be a determining factor in both the rate of TLR agonist influx from the gut and NAFLD/NASH progression. In comparison with those without SIBO, patients with SIBO had significantly higher endotoxin levels and higher CD14 mRNA, nuclear factor kappa B mRNA and TLR4 protein expression [45]. Patients with NASH had significantly higher endotoxin levels and more intense hepatic TLR4 protein expression than patients without NASH [45]. Although LPS loading resulted in hepatic inflammation in both maintenance-food- and high-caloric-diet-fed mice, high-caloric-diet feeding was more crucially important in the progression of NAFLD during the triggering phase [46].

The bile acids, gut microbiota and metabolome that contribute to the regulation of glucose and lipid metabolism were affected by the Myd88 level in hepatocytes, resulting in altered susceptibility to obesity, such as induction of type 2 diabetes and NASH, in both mice and humans [47]. Lactobacillus paracasei attenuated hepatic steatosis by M2-dominant Kupffer cell polarization in a NASH mouse model [48].

Patients with NASH were shown to have a lower percentage of Bacteroidetes (Bacteroidetes to total bacteria counts) than those with simple steatosis or living liver donors (representing healthy controls) [49]. The inverse and diet-/BMI-independent association between the presence of NASH and the percentage of Bacteroidetes in the stool also supports the notion that the intestinal microbiota may play a role in the development of NAFLD [49].

Yuan et al. reported that a greater abundance of Gram-negative bacteria in the gut microbiome was observed in obese and NASH patients than in normal subjects [50]. Among NASH patients, serum endotoxin is not correlated with disease severity, suggesting that endotoxemia is not required in the pathogenesis [49]. Subjects with NAFLD had slightly higher lipopolysaccharide-binding protein (LBP) (P < 0.001) and EndoCab immunoglobulin G (IgG) (*P* = 0.013) levels than those without NAFLD [51]. These endotoxin markers are associated with NAFLD in the general population but do not have a major effect on NASH and/or fibrosis. People with modest alcohol consumption may have lower serum endotoxin levels than those with heavy alcohol consumption.

A meta-analysis of four randomized trials involving 134 NAFLD and NASH patients revealed that probiotic therapies can reduce AST, ALT, total cholesterol and TNF-α, and improve insulin resistance in these patients [52]. Modulation of the gut microbiota may afford a novel treatment for NAFLD [52]. Mosapride citrate, a gastroprokinetic agent, showed a protective effect against MCD diet-induced NASH development in a mouse model, with a possible involvement of increased fecal lactic acid bacteria, protection against colon inflammation and an elevated plasma-glucagon-like peptide-1 (GLP-1) concentration [53].

Nakano et al. observed that a history of appendectomy was significantly more frequent in NAFLD patients with advanced fibrosis than in those without fibrosis [54]. Appendectomy may thus be one of the risk factors for advanced fibrosis in NAFLD [54]. In adult patients with NAFLD, dysbiosis is associated with altered bile acid homeostasis, which renders patients at increased risk of hepatic injury [55]. Gut microbiota alterations are closely correlated with dysregulated bile acid levels in the liver and feces of streptozotocin- and high-fat-diet-induced NASH–HCC mice [56]. Calcium reduces liver injury in mice on a high-fat diet, as well as inducing alterations in the microbial and bile acid profiles [57]. Intestinal bacteria may play an important role in the development of NAFLD and NASH [58,59,60,61,62,63,64,65], and they may constitute a therapeutic target. Further studies will be needed to determine the therapeutic utility [66,67,68,69].

## 4. NASH and HCC

Among 98 patients over the age of 18 with pathologically confirmed HCC in a liver transplantation center in the United States, 50%, 23% and 10% of the etiologies of cirrhosis were chronic HCV infection, ALD and NASH, respectively [70]. NAFLD is now considered the third most common cause of HCC in the United States [71]. In the United States, 54.9%, 16.4%, 14.1% and 9.5% of HCC cases are related to HCV, ALD, NAFLD and HBV, respectively [72]. The incidence of NAFLD-related HCC is increasing at a 9% annual rate [71,72]. In approximately 13% of HCC patients reported in a study conducted by the Veterans Administration, 80% and ~13% of the patients did or did not have cirrhosis, respectively [73]. Having NAFLD or metabolic syndrome was independently associated with HCC in the absence of cirrhosis [73].

ALD (19.9%) and NASH (6.3%) are reportedly contributing etiologies of liver cirrhosis in Japan, with viral hepatitis, particularly HCV-related hepatitis (48.2%), being the major cause [74]. Among the non-viral etiologies in Japan, ALD and NASH-related liver cirrhosis exhibited a notable increase in prevalence from diagnosis before 2008 to in and after 2014 (from 13.7% to 24.9% and from 2.0% to 9.1%, respectively). The 20th nationwide Follow-up Survey of Primary Liver Cancer in Japan over the two-year period from January 1, 2008, to December 31, 2009, found HBsAg-positive and anti-HCV-positive HCC occurred in 2768 of 18,219 (15.2%) patients and 10,976 of 18,097 (60.7%) patients, respectively, suggesting that the non-B non-C cases were ~20% [75]. In Japan, the five-year HCC rates of NASH cirrhosis and HCV cirrhosis were 11.3% and 30.5%, respectively [76]. HCC was the leading cause of death in both the NASH cirrhosis and HCV cirrhosis groups (47% and 68%, respectively) [76].

In Korea, cryptogenic HCC accounts for approximately 7% of all HCC cases. Cryptogenic HCC includes features of older age at diagnosis, more frequent occurrence of metabolic syndrome and less aggressive tumor characteristics. Cryptogenic HCC may be a form of “burnt-out” NAFLD [77]. The contribution of NASH to HCC in Asian countries is lower (7%–16%) than in the West [78].

By 2010, NAFLD accounted for 41/118 (34.8%) HCC cases in England [79]. Irrespective of the associated etiologies, metabolic risk factors were present in 78/118 (66.1%) cases in 2010 [79]. Large-scale epidemiological studies have repeatedly associated obesity and type 2 diabetes mellitus with a risk for HCC [80]. At diagnosis, patients with NAFLD-associated HCC are older, have more extrahepatic comorbidities and have a higher prevalence of non-cirrhosis (~30%) than patients with HCC from other causes [80]. Yasui et al. reported that most patients with NASH who develop HCC are men, and most of the patients have obesity, diabetes and hypertension [81]. Men with NASH appear to develop HCC at a less advanced stage of liver fibrosis than women with NASH.

## 5. Genetic Factors in Patients with NAFLD, NASH and HCC

Caldwell et al. reported that, among 206 cryptogenic cirrhosis patients, only two patients (1%) were of African American descent, whereas 195 patients (95%) were of European American descent [82]. Although the prevalence of diabetes mellitus was similar, the prevalence of cryptogenic cirrhosis among Hispanic and African American patients was 3.1-fold higher and 3.9-fold lower, respectively, than that among European American patients [83]. Thus, the progression of NAFLD to cirrhosis and HCC may differ in different ethnic groups [84].

### 5.1. Patatin-Like Phospholipase-3 (PNPLA3)

To identify DNA sequence variations that contribute to interindividual differences in NAFLD, Romero et al. performed a genome-wide association study of nonsynonymous sequence variations (n = 9229) in a multiethnic population [85]. One PNPLA3 variant (rs738409; I148M) was significantly associated with increased hepatic fat levels (*P* = 5.9×10^−10^) and with hepatic inflammation (*P* = 3.7×10^−4^) [84]. This variant was most common in Hispanics, who are most susceptible to NAFLD. The hepatic fat content was > twofold higher in patients with homozygous PNPLA3-148M than in noncarriers [85]. PNPLA3 encodes a 481-amino-acid protein of unknown function that belongs to the patatin-like phospholipase family [86].

The identification of a second allele of PNPLA3 (i.e., S453I) that was independently associated with hepatic fat content further supports a role for PNPLA3 in determining hepatic triglyceride levels [85]. Yuan et al. also reported that the PNPLA3 variant (rs2281135, T) is associated with the plasma levels of certain liver enzymes [87]. Obese Southern Europeans carrying the PNPLA3-148M allele have increased indices of liver damage independent of proxy measures of insulin resistance [88]. Kawaguchi et al. also found that Matteoni type 4 NAFLD is both a genetically and clinically different subset from the other spectra of the disease and that the PNPLA3 variant is strongly associated with the progression of NASH in Japanese patients with NAFLD [89,90].

Singal et al. performed a meta-analysis of nine studies with 2937 patients and found that PNPLA3 was associated with an increased risk of HCC in patients with cirrhosis (OR 1.40, 95% CI 1.12–1.75) [91]. Upon subgroup analysis, PNPLA3 was found to be an independent risk factor for HCC in patients with NASH or ALD-related cirrhosis (OR 1.67, 95% CI 1.27-2.21). Liu et al. also reported that bearing the PNPLA3 rs738409 c.444C>G minor allele (encoding the I148M variant) was associated not only with a greater risk of progressive steatohepatitis and fibrosis but also with HCC in patients with NAFLD [92]. Ueyama et al. revealed that the PNPLA3 G allele and JAZF zinc-finger 1 (JAZF1) rs864745 G allele are associated with non-HBV and non-HCV-related HCC in Japanese patients with type 2 diabetes mellitus [93].

### 5.2. Hydroxysteroid 17-Beta Dehydrogenase 13 (HSD17B13)

NAFLD is characterized by a massive accumulation of lipid droplets (LDs) [94]. Su et al. identified 17β-HSD13 as a pathogenic protein involved in the development of NAFLD [94]. One splice variant (rs72613567: TA) in HSD17B13 that encodes the hepatic lipid droplet protein hydroxysteroid 17-beta dehydrogenase 13 was found to be associated with reduced levels of ALT and AST [95]. This variant was associated with a reduced risk of NAFLD (by 17% (95% CI, 8 to 25) in heterozygotes and by 30% (95% CI, 13 to 43) in homozygotes) and nonalcoholic cirrhosis (by 26% (95% CI, 7 to 40) in heterozygotes and by 49% (95% CI, 15 to 69) in homozygotes) [95].

Chen et al. reported that lower HSD17B13 in peritumoral tissues was associated with worse recurrence-free survival and overall survival in HCC patients, although they mainly studied viral-hepatitis-associated HCC [96]. They also showed that HSD17B13 delayed G1/S progression in HCC cells [96]. Patients carrying rs72613567, a splice variant with an adenine insertion (A-INS), exhibited HSD17B13 levels that decreased proportionally with allele dosage [97]. Whole-transcriptome genotype profiling also showed an overrepresentation of immune-response-related pathways. The HSD17B13 rs72613567 A-INS allele reduces the risk of NASH and progressive liver damage [97].

HSD17B13 has retinol dehydrogenase (RDH) activity, with this enzymatic activity dependent on lipid droplet targeting and cofactor binding sites [98]. May et al. reported an association of variants of HSD17B13 with certain specific features of NAFLD histology and identified the enzyme as a lipid droplet associated RDH [98]. The HSD17B13 minor allele rs6834314 was significantly associated with increased steatosis but decreased inflammation, ballooning, Mallory–Denk bodies and liver enzyme levels in 768 adult Caucasians with biopsy-proven NAFLD, as well as cirrhosis, in the general population [98]. Yang et al. reported that the HSD17B13 rs72613567 loss of function variant protected against HCC development in patients with ALD [99].

### 5.3. Other Genetic Variants

Namjou et al. reported that three SNPs in the PNPLA3-SAMM50 region, rs738409, rs738408 and rs3747207, displayed the strongest association with NAFLD [100]. They also identified certain novel loci for NAFLD disease severity, including one locus near IL17RA (rs5748926, *p* = 3.80 × 10^− 8^) that was associated with the NAS score, and another near ZFP90-CDH1 that was associated with fibrosis (rs698718, *p* = 2.74 × 10^− 11^) [100].

Klotho beta (KLB) mediates the binding of fibroblast growth factor 21 (FGF21) to the FGF receptor (FGFR). FGF21-KLB-FGFR signaling plays an important role in multiple hepatic metabolic systems [101]. Ji et al. reported that KLB SNPs were related to obesity and hepatic inflammation, which may be involved in the pathogenesis of NAFLD [101]. Xu et al. reported that the TM6SF2 E167K variant was associated with NAFLD in Northeast China [102]. A meta-analysis showed a correlation between the rs1501299 polymorphism in the adiponectin gene and the risk of NAFLD [103]. Kawaguchi et al. found the rs17007417 in dysferlin (DYSF) (*p* = 5.2 × 10^−7^, OR (95%CI) = 2.74 (1.84–4.06)) to be a SNP associated with NASH–HCC [104]. Rs17007417 in DYSF was significantly different in NASH–HCC cases compared with controls, as well any of the four Matteoni subgroups [90]. Further studies will be needed to determine the effects of these genetic variants on hepatocarcinogenesis.

## 6. Molecular Pathways in Patients with NASH and HCC

We recently reviewed the molecular mechanisms driving the progression of liver cirrhosis toward HCC in chronic HBV and HCV infection [105] and found that exosomes were involved in hepatocarcinogenesis [106]. We expect similar processes to be involved in hepatocarcinogenesis in patients with NASH. Specific microRNA expression patterns may also permit NAFLD progression into HCC [107]. Guo et al. reported that miR-301a-3p, which was found to increase as the liver condition progressed from normal to cirrhosis in the trend analysis, is upregulated in HCC [107]. The Cancer Genome Atlas (TCGA) data show that the expression of miR-301a increases and that of miR-375 decreases as HCC progresses from the early to late stages [107].

Sun et al. reported that an androgen receptor (AR)-driven oncogene, cell cycle-related kinase (CCRK), interacts with obesity-induced pro-inflammatory signaling to promote NASH-related hepatocarcinogenesis. Activation of the mTORC1/4E-BP1/S6K/SREBP1 cascades via GSK3β phosphorylation and the STAT3-AR-CCRK-mTORC1 pathways is involved in this process [108]. STAT3 and AR signaling are also involved in hepatocarcinogenesis in patients infected with HCV [109]. The dual activities of the inflammatory-CCRK circuit in driving metabolic and immunosuppressive reprogramming through mTORC1 activation play a role in forming a protumorigenic microenvironment for HCC development [108]. Oxidative stress also promotes pathological polyploidization, which is an early event in NAFLD and may contribute to HCC development [110].

Altered patterns of NASH-specific and NASH-related HCC-specific DNA methylation that were not evident in virus-infected noncancerous samples or 37 samples of HCC associated with either HBV or HCV infection were observed in certain tumor-related genes, such as nuclear receptor binding SET domain protein 2 (NSD2/WHSC1), and are frequently associated with mRNA expression abnormalities [111]. Some DNA methylation alterations during nonalcoholic steatohepatitis (NASH)-related hepatocarcinogenesis appear to be different from those of viral hepatitis-related hepatocarcinogenesis [111]. Among the genes displaying DNA hypomethylation during NASH-related carcinogenesis is MAML3. MAML3 is a critical transcriptional co-activator in the Notch signaling pathway and is also a co-activator of β-catenin-mediated transcription [111]. MAML3 increases the transcriptional activity of β-catenin. 

Stochastic models of the genome-wide genetic and epigenetic network (GEN) in human cells based on molecular mechanisms, including transcription factor regulation, miRNA repression, DNA methylation and protein–protein interactions (PPIs) have shown that the hepatocarcinogenesis associated with NAFLD and NASH is induced by DNA methylation of histone cluster 2 H2B family member E (HIST2H2BE), heat shock protein family B (small) member 1 (HSPB1), ribosomal protein L30 (RPL30), aldolase, fructose-bisphosphate B (ALDOB), and the regulation of miR-21 and miR-122 [112]. Genetic and epigenetic changes in the progression of NASH to HCC are shown in Table 2.

## 7. Conclusions

We described the molecular mechanisms of the progression to HCC in patients with NAFLD/NASH (Figure 1). A recent real-world study of 18 million patients in four European cohorts demonstrated that the strongest independent predictor of a diagnosis of HCC or cirrhosis was a baseline diagnosis of diabetes [113]. A recent review reports that alterations in lipid-related genes are important for the progression of NASH into HCC [114]. In NAFLD/NASH-associated HCC, including its epidemiology, the key feature is the hepatic NAFLD/NASH microenvironment that promotes hepatocarcinogenesis [115]. Further analysis of the mechanisms that lead to HCC should be urgently explored to find new therapeutic targets, as well as biomarkers for the early detection of HCC in patients with NAFLD/NASH.

## Figures and Tables

**Figure 1 ijms-21-01525-f001:**
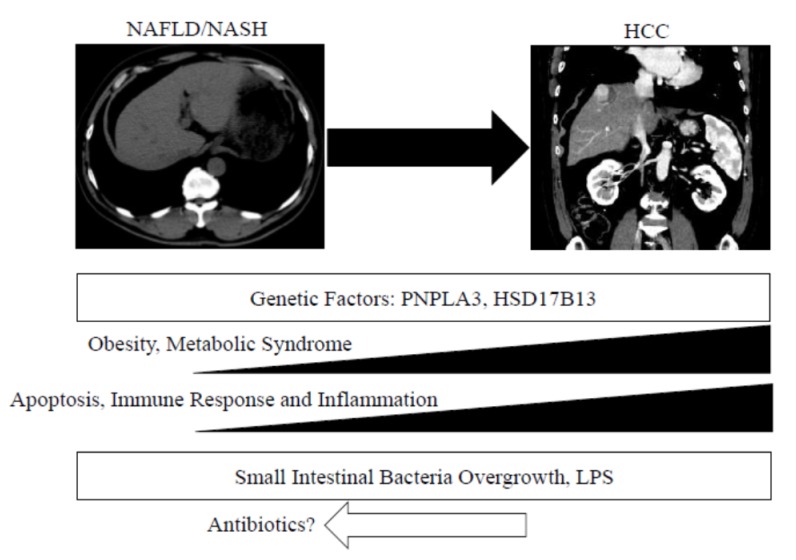
Molecular mechanisms: the connections between nonalcoholic fatty liver disease, steatohepatitis and hepatocellular carcinoma. NAFLD, nonalcoholic fatty liver disease; NASH, nonalcoholic steatohepatitis; HCC, hepatocellular carcinoma; PNPLA3, patatin-like phospholipase-3; HSD17B13, hydroxysteroid 17-beta dehydrogenase 13; LPS, lipopolysaccharide.

**Table 1 ijms-21-01525-t001:** Contribution factors in the pathogenesis of NAFLD and NASH.

Factors	Observation
Obesity	Hyperinsulinemia, Insulin resistance
Apoptosis	ER stress, Oxidative stress
Immune and inflammatory pathways	Activation of macrophages, Iron metabolism,
Intestinal bacteria	SIBO, Intestinal mucosa barrier malfunction, Intestinal microbiota
SNPs	PNPLA3, HSD17B13
Epigenetic alterations	MicroRNA, DNA methylation

NAFLD, nonalcoholic fatty liver disease; NASH, nonalcoholic steatohepatitis; ER, endoplasmic reticulum; SIBO, small intestinal bacterial overgrowth; SNP, Single nucleotide polymorphism; PNPLA3, Patatin-like phospholipase-3; HSD17B13, Hydroxysteroid 17-beta dehydrogenase 13.

**Table 2 ijms-21-01525-t002:** Genetic and epigenetic changes in the progression of NASH to HCC.

Molecules
miR-301a-3p, miR-375 [107]
CCRK [108]
MAML3 [111]
DNA methylation of HIST2H2BE, HSPB1, RPL30, ALDOB [112]
miR-21, miR-112 [112]

NASH, nonalcoholic steatohepatitis; HCC, hepatocellular carcinoma; CCRK, cell cycle-related kinase; HIST2H2BE, histone cluster 2 H2B family member E; HSBP1, heat shock protein family B (small) member 1; RPL30, ribosomal protein L30; ALDOB, aldolase, fructose-bisphosphate B.

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
