# Peer review of "Molecular Mechanisms: Connections between Nonalcoholic Fatty Liver Disease, Steatohepatitis and Hepatocellular Carcinoma"

_ijms, 2020, doi:10.3390/ijms21041525_

Round 1

Reviewer 1 Report

The quality of this manuscript has been largely improved and it is acceptable to be published now.

Author Response

Thank you very much for your encouraging comments.

Reviewer 2 Report

Comments

The revised review has improved but is still difficult to read.

For example the paragraph “It is well known that obesity and metabolic syndrome are associated with NAFLD and NASH. Significant weight loss can lead to the improvement of NAFLD and NASH [11]. Hepatocyte apoptosis plays a central role in the activation and development of NAFLD and NASH [12]. Immune and inflammatory pathways also play a role in the pathogenesis of NAFLD and NASH [13]” can be simplified: Obesity, metabolic syndrome and hepatocyte apoptosis are associated with NAFLD and NASH.

Another example is the sentence “Among 98 patients over the age of 18 with pathologically confirmed hepatocellular carcinoma who had a liver transplantation between January 1, 2012, and December 31, 2017, at the University of Massachusetts, MA, United States, 50%, 23% and 10% of the etiologies of cirrhosis were chronic HCV infection, ALD and NASH, respectively [70]”. A lot of information is provided but is it necessary to mention that transplantations took place at the University of Massachusetts between January 1st, 2012, and December 31st, 2017?

Unnecessary information should be deleted. Extensive proofreading and streamlining is still recommended.

Author Response

Answer to Reviewer 2’s comment 1: ”The revised review has improved but is still difficult to read. For example the paragraph “It is well known that obesity and metabolic syndrome are associated with NAFLD and NASH. Significant weight loss can lead to the improvement of NAFLD and NASH [11]. Hepatocyte apoptosis plays a central role in the activation and development of NAFLD and NASH [12]. Immune and inflammatory pathways also play a role in the pathogenesis of NAFLD and NASH [13]” can be simplified: Obesity, metabolic syndrome and hepatocyte apoptosis are associated with NAFLD and NASH..

Thank you for your valuable comments. According to your suggestion, we revised our manuscript as follows.

In page 2, lines 65-66 of the revised manuscript,

3. Pathophysiology of NASH

Obesity, metabolic syndrome and hepatocyte apoptosis are associated with NAFLD and NASH [11-13]. Both innate and acquired immunity mediate the development of insulin resistance and NASH [14]. Recent studies revealed that intestinal bacteria are also involved in the development of NASH [15-17]. Thus, many factors contribute to the development of NAFLD and NASH (Table 1).

Answer to Reviewer 2’s comment 2: ”Another example is the sentence “Among 98 patients over the age of 18 with pathologically confirmed hepatocellular carcinoma who had a liver transplantation between January 1, 2012, and December 31, 2017, at the University of Massachusetts, MA, United States, 50%, 23% and 10% of the etiologies of cirrhosis were chronic HCV infection, ALD and NASH, respectively [70]”. A lot of information is provided but is it necessary to mention that transplantations took place at the University of Massachusetts between January 1st, 2012, and December 31st, 2017?

Unnecessary information should be deleted. Extensive proofreading and streamlining is still recommended.”

Thank you for your valuable comments. According to your suggestion, we revised our manuscript as follows.

In page 5, lines 230-231 of the revised manuscript,

4. NASH and HCC

              Among 98 patients over the age of 18 with pathologically confirmed HCC in the liver transplantation center of the United States, 50%, 23% and 10% of the etiologies of cirrhosis were chronic HCV infection, ALD and NASH, respectively [70]. 

This manuscript is a resubmission of an earlier submission. The following is a list of the peer review reports and author responses from that submission.

Round 1

Reviewer 1 Report

Authors reviewed the papers and described current advances associated with the molecular mechanism contributing to the progression of hepatic fibrosis and HCC in patients subjected to NAFLD/NASH. The manuscript raises several interesting points of view concerning the development of hepatic carcinoma. However, there are still several important issues should be mentioned in this manuscript.

First of all, the present manuscript is just a pale sequel of the previous document. Lack of novel points of view in this field. There is little description on the molecular mechanisms related to the correlations between NAFLD/NASH and HCC. The authors should provide easily read tables rather text narration to enhance the value of this manuscript.

Reviewer 2 Report

Summary

The present review summarizes mechanisms implicated in NAFLD-NASH-HCC transition.

Comments

The review is difficult to read and to understand. A lot of information is provided but not logically connected. Extensive proofreading and streamlining is necessary. Several reviews on the topic have been published recently. Some of them cover all the information in the present review (e.g. Anstee, Q.M., Reeves, H.L., Kotsiliti, E. et al.From NASH to HCC: current concepts and future challenges. Nat Rev Gastroenterol Hepatol 16, 411–428 (2019). https://doi.org/10.1038/s41575-019-0145-7).